

# A laboratory comparison of the interactions between three plastic mulch types and 38 active substances found in pesticides

Nicolas Beriot[1,2], Paul Zomer[3], Raul Zornoza[2] and Violette Geissen[1]

[1] Soil Physics and Land Management Group, Wageningen University and Research, Wageningen, Netherlands
[2] Sustainable Use, Management and Reclamation of Soil and Water Research Group, Universidad Politécnica de Cartagena, Cartagena, Murcia, Spain
[3] Wageningen Food Safety Research, Wageningen University and Research, Wageningen, Netherlands

## ABSTRACT

**Background:** In semi-arid regions, the use of plastic mulch and pesticides in conventional agriculture is nearly ubiquitous. Although the sorption of pesticides on Low Density Polyethylene (LDPE) has been previously studied, no data are available for other plastics such as Pro-oxidant Additive Containing (PAC) plastics or "biodegradable" (Bio) plastics. The aim of this research was to measure the sorption pattern of active substances from pesticides on LDPE, PAC and Bio plastic mulches and to compare the decay of the active substances in the presence and absence of plastic debris.

**Methods:** For this purpose, 38 active substances from 17 insecticides, 15 fungicides and six herbicides commonly applied with plastic mulching in South-east Spain were incubated with a $3 \times 3$ cm$^2$ piece of plastic mulch (LDPE, PAC and Bio). The incubation was done in a solution of 10% acetonitrile and 90% distilled water at 35 °C for 15 days in the dark. The Quick Easy Cheap Effective Rugged Safe approach was adapted to extract the pesticides.

**Results:** The sorption behavior depended on both the pesticide and the plastic mulch type. On average, the sorption percentage was ~23% on LDPE and PAC and ~50% on Bio. The decay of active substances in the presence of plastic was ~30% lesser than the decay of active substances in solution alone. This study is the first attempt at assessing the behavior of a diversity of plastic mulches and pesticides to further define research needs.

Corresponding author
Nicolas Beriot, nicolas.beriot@wur.nl

## INTRODUCTION

The use of plastic mulching has become a well-established technique to increase the profitability of many crops (*Kasirajan & Ngouajio, 2012*). The European Commission estimated in 2016 that 100,000 tons of plastic mulch is used per year in the European

Union (*European Commission, 2016*). Plastic mulch is generally used for one or all of the following three reasons: decreasing evaporation, decreasing weed competition or increasing soil temperature (*Steinmetz et al., 2016*). After crop harvest, some farmers try to remove the plastic mulch but debris is left in the soil. Other farmers simply incorporate the plastic into the soil (*Kasirajan & Ngouajio, 2012*). Once the plastic is in the environment, the low degradation rate of plastic debris facilitates its accumulation (*Rillig, 2012*).

The plastic mulch degradation process can be explained by looking at three main underlying factors: abiotic conditions, microbial requirements and properties of the plastic mulch material (*Hayes et al., 2012*). The most common plastic used for mulching is Low Density Polyethylene (LDPE) (*Kasirajan & Ngouajio, 2012*). LDPE is a fully saturated polymer of hydrocarbons which makes it highly resistant (*Crawford et al., 2017a*). Consequently, LDPE mulch needs to be removed after harvest and LDPE debris accumulates in the environment. Some plastic producers have tried to improve the degradation processes of plastic to avoid plastic mulch removal and plastic debris accumulation. Pro-oxidant Additive Containing (PAC) plastics are polymers, mainly LDPE, which contain a pro-oxidant additive to enhance oxidation and photo-degradation (*Selke et al., 2015*). In the presence of light and under aerobic conditions, PAC plastics degrade quickly into small pieces. Small fragmented debris is more likely to be further degraded by microorganisms (*Ahmed et al., 2018*). PAC plastics are also known as "oxo-degradable" or "oxo-biodegradable" (*Hogg et al., 2016*). However, when incorporated into the soil, the degradation process is minimized due to the absence of UV-light (*Hogg et al., 2016*) and PAC debris may accumulate. Over the last few years, new mulching films that can be degraded by microorganisms in the soil have been developed (*Hayes et al., 2017*; *Sintim & Flury, 2017*). They are usually sold as "biodegradable" (Bio) mulch (*Van den Oever et al., 2017*). Biodegradable mulch can be made of a diversity of polymers (*Kijchavengkul & Auras, 2008*) either biobased, synthetic or a blend of both. Biodegradation of polymeric mulch films relies on three fundamental steps: the colonization of the polymer surfaces by soil microorganisms, the enzymatic depolymerization of the polymer by extracellular hydrolases secreted by the colonizing microorganisms and the microbial utilization of the hydrolysis products that are released from the polymer (*Sander, 2019*). Therefore, a larger contact area helps colonization and polymers containing functional groups that can be enzymatically hydrolyzed increase the degradation rate. About 3,000 tons of biodegradable plastic mulch are used each year in the European Union (*European Commission, 2016*). In order to properly manufacture plastic mulches, additives such as nucleating agents, plasticizers, performance additives and lubricants are required (*Briassoulis, 2004*; *Hayes et al., 2012*; *Shen, Worrell & Patel, 2010*). It is important to note that manufacturers do not normally share the chemical structures of the raw materials or the additives that are used in plastic production in order to protect their products from being duplicated.

In arid and semi-arid areas, where water deficits are common, the use of plastic mulching for irrigated crops is widespread. It is a technically and economically feasible strategy used to prevent evaporation and reduce water consumption. This is the case in regions such as the Loess plateau in China (*Jiang et al., 2016*) and in the Murcia Region of South-eastern Spain (*Van der Meulen, Nol & Cammeraat, 2006*). In addition to plastic
mulch, pesticides are used in conventional agriculture to control weeds, insects and fungi (*Van den Oever et al., 2017*). The synergetic effect of plastic debris and pesticide residues on degradation and on the terrestrial environment is not sufficiently understood. *Nerín et al. (1996)* and *Sharom & Solomon (1981)* studied the sorption rates of nine different active substances in pesticides on LDPE and found a sorption rate between 20% and 100% after 15 days at 35 °C. Adsorbed active substances are less likely to be degraded (*Kasirajan & Ngouajio, 2012*) and may be released when ingested by an organism (*Teuten et al., 2007*). Furthermore, microplastics may be carriers for pesticide residues when transported through the terrestrial environment. The modification of the degradation patterns of active substances might affect the soil organism community due to the toxicity of the active substances. Moreover, the microbial activity plays a major role in Bio plastic degradation. Therefore, adsorption of active substances could potentially decrease plastic debris degradation.

Previous studies determined the occurrence of adsorbed organic contaminants on different plastic polymers (*Crawford et al., 2017b*; *Hirai et al., 2011*; *Mato et al., 2001*) and have characterized the sorption of different organic contaminants on plastics (*Lee, Shim & Kwon, 2014*; *Liu et al., 2019*; *Mato et al., 2001*; *Ramos et al., 2015*; *Seidensticker et al., 2018*; *Teuten et al., 2007*). Most studies were focused on the aquatic environment and coastal areas but *Hüffer et al. (2019)* showed that the sorption on polyethylene microplastics influenced the transport of hydrophobic organic pollutants in soils. More data are required to assess the sorption of commonly used pesticides on LDPE and on new types of plastic.

As a preliminary investigation to address these data gaps, the sorption of 38 active substances from 17 insecticides, 15 fungicides and six herbicides commonly used along with plastic mulching in South-eastern Spain, were tested on three types of plastic mulch: LDPE, PAC and Bio. The objectives of this research were to measure the sorption of a mixture of 38 active substances on plastic mulch and to compare the decay of adsorbed and non-adsorbed active substances. We hypothesized that sorption rates would be different for each specific active substance and each specific plastic. For example, the Bio mulch was believed to be the most prone to sorb active substances (*Boivin, Cherrier & Schiavon, 2005*; *Crawford et al., 2017a*). Furthermore, we hypothesized that sorption would reduce the degradation of active substances (*Nerín et al., 1996*; *Ramos et al., 2015*).

## MATERIALS AND METHODS

A laboratory single point sorption experiment was set up to test the sorption of a mixture of active substances on plastic mulches. Previously, eight vegetable farmers in the region of Murcia (Southeast Spain) were interviewed to discover which types of pesticides and plastic mulches were commonly used in the research area. All interviewed farmers used either LDPE, PAC or Bio plastic mulch in their crop production. All farmers used similar vegetable rotations and the type of plastic mulch used was not linked to the type of crop. We were able to assemble a full list of the active substances in the pesticides that were used by the farmers. Some active substances on the list were not analyzed due to logistical

**Table 1 Type of plastic mulch and incubation solution for the seven treatments.**

| Treatment | Plastic mulch | Incubation solution (10% acetonitrile and 90% $H_2O$) |
| --- | --- | --- |
| LDPE+P | LDPE | Active substances mixture |
| PAC+P | PAC | Active substances mixture |
| Bio+P | Bio | Active substances mixture |
| LDPE+W | LDPE | No-active substances |
| PAC+W | PAC | No-active substances |
| Bio+W | Bio | No-active substances |
| P | – | Active substances mixture |

Note:
LDPE, Low Density Polyethylene mulch; PAC, Pro-oxidant Additive Containing mulch; Bio, Biodegradable mulch; P, mixture containing the active substances; W, mixture without the active substances.

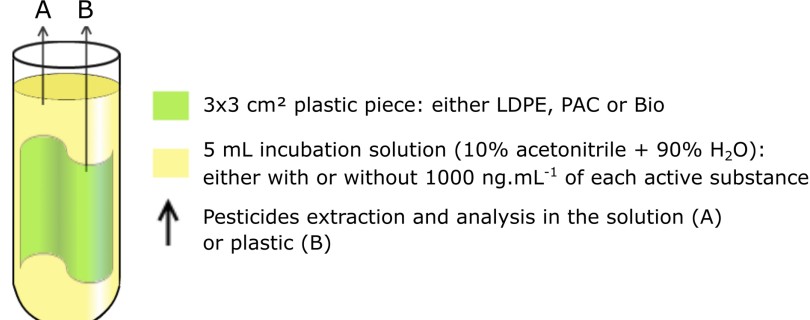

**Figure 1 Glass tube set up drawing.** LDPE, Low Density Polyethylene mulch; PAC, Pro-oxidant Additive Containing mulch; Bio, Biodegradable mulch. Pesticides content was analyzed in the solution (A) and in the plastic (B). All experiments were performed in duplicate. Glass tubes were incubated at 35 °C for 15 days.               

and financial limitations. The final list of 38 active substances from 17 insecticides, 15 fungicides and six herbicides is presented in Table S3.

Three plastic mulches: LDPE, PAC and Bio, were incubated with or without active substances. Additionally, one control treatment containing a mixture of the active substances without plastic was also tested. In total, seven treatments were set up in glass tubes (Table 1). All treatments were carried out in duplicate. Each tube contained five mL of a solution (either with or without the mixture of active substances) and a piece of $3 \times 3$ cm$^2$ plastic mulch, depending on the treatment (Fig. 1). Therefore, we can distinguish two phases: the plastic mulch and the incubation solution.

## Plastic mulch types used in the experiment

Samples of unused plastic mulch were collected from farmers' warehouses located in the region of Murcia for each of the mulches: LDPE, PAC and Bio. All three types of mulches were black. The detailed composition of the plastic was not given by the producers since it was protected by intellectual property regulations. In addition to the main polymer, all plastics contained additives used to control the color, elasticity and resistance of the mulch (*Crawford et al., 2017a*; *Sintim & Flury, 2017*). The LDPE mulch came from

Reyenvas (Spain). LDPE plastic mulch is designed to be resistant and removed after the harvest. The PAC mulch (commercial name "actiblack") came from Trioplast SMS SAS (France). PAC mulch is made of LDPE with the addition of a pro-oxidant additive that increases its decay such that farmers usually incorporate it into the soil after harvest instead of removing it. Finally, Bio mulch (commercial name "Sotrafilm Black Biodegradable") was bought from Sotrafa (Spain). The available information states that it is a "biopolymer film made with biodegradable and renewable raw materials and particular carbon black content to get an optimum opacity for mulching use" (*Sotrafa, 2018*). The compliance with the biodegradable plastic mulch norms EN 17033:2018 (*CEN, 2018*) or ISO 17556:2019 (*ISO, 2019*) was not specified. The composition of the biodegradable mulch was investigated using the Varian 1000 FTIR (Fourier transform infrared) spectrometer from the Aquatic Ecology and Water Quality Management group of Wageningen University. Eight scans were performed for the background and the samples. The spectrometer produced spectra ranging from $3750 \, cm^{-1}$ to $400 \, cm^{-1}$ with a resolution of $4 \, cm^{-1}$. The comparison of the spectra with polymer libraries (HR Hummel Polymer and Additives, HR Spectra Polymers and Plasticizers by ATR, HR Sprouse Polymers by Transmission) gave high percentages of match for Polyester terephthalic acid (78% match), Polybutylene terephthalate (72% match) and polyethylene terephthalate (65.6% match) (Fig. S1; Table S1). Therefore, the biodegradable plastic mulch may have been composed of Polybutylene terephthalate, polyethylene terephthalate or other similar copolyester of terephthalic acid. It is most likely that the main polymer was Polybutylene adipate terephthalate as it is a copolyester of terephthalic acid commonly used for its biodegradability properties (*Weng et al., 2013*).

The pieces of plastic mulch that were collected were manually cut into $3 \times 3 \, cm^2$ pieces before incubation. The $3 \times 3 \, cm^2$ pieces of mulch were manipulated so that they fit into the glass tubes but they were not folded (Fig. 1). Plastic pieces were fully immersed in the solution so that the incubation solution was in contact with the whole surface of the piece of plastic.

## Incubation solution and incubation conditions

For each of the 38 active substances (Table S3), the Pesticide Properties Database (*BPDB, 2018*; *PPDB, 2018*) was used to get the molar mass, the aqueous hydrolysis half-life time at 20 °C and pH 7 (DT50; indicator of degradation in water), the solubility in water at 20 °C and the octanol-water partition coefficient at pH 7, 20 °C (log P). The octanol-water partition coefficient (log P) was used as a measure of the active substances hydrophobicity, which plays a key role in sorption (*Leo, Hansch & Elkins, 1971*). A concentration of $1,000 \, ng \cdot mL^{-1}$ of each active substance was mixed in a solution of 10% acetonitrile and 90% distilled water so that there was 5,000 ng of each active substance in the final volume of five mL. The concentration was the same as in *Nerín et al. (1996)* and the mass of pesticides available per area of plastic was similar. Acetonitrile in the incubation solution may have helped the dissolution of hydrophobic active substances since the solubility in water for some of the substances was low (Table S3). The initial presence of active substances in the plastic was assessed using the same incubation solution (90% distilled water + 10% acetonitrile) without active substances applied (Treatments LDPE+W, PAC+W, Bio+W).
All glass tubes were incubated at 35 °C for 15 days in a laboratory oven. The temperature was representative of the temperature under the plastic mulch in semi-arid regions (*Nerín et al., 1996*). Tubes were kept in the dark, without additional stirring or oxygenation during the 15 days. A period of 15 days was enough time to reach the sorption equilibrium, as reported on LDPE films by *Nerín et al. (1996)* and allowed us to study the degradation of the substances.

## Active substance extraction and determination

After incubation, the plastic pieces were carefully washed with distilled water, cut into $5 \times 5$ mm$^2$ pieces and transferred to a 50 mL tube for active substance extraction. The extraction method was adapted from *Nerín et al. (1996)* and the Quick Easy Cheap Effective Rugged Safe approach (*Anastassiades et al., 2003*). *Nerín et al. (1996)* showed with a similar extraction method that a single extraction was sufficient for a quantification. The analytical method was similar to the one described in *Mol et al. (2008)* and *Silva et al. (2018a)*. Plastic tubes and plastic vials were used for extraction and quantification procedures. Given the concentration of acetonitrile and the short time of extraction, we do not expect significant losses of active substances based on the quality assessment method of *Mol et al. (2008)* and *Silva et al. (2018a)*. All plastic samples were spiked with $^{13}$C-caffeine (used as internal standard to assess the procedure efficiency), and mixed with five mL of distilled water and 10 mL of acetonitrile containing 1% acetic acid. Tubes were exposed to an ultrasonic bath for 1 h and agitated end-over-end for another hour. Then, one g of sodium acetate and four g of magnesium sulfate were added to induce phase separation. After centrifugation, 250 µL of the supernatant (acetonitrile phase) was collected, mixed with 250 µL of distilled water and filtered in a filter vial for analysis. The incubation solution was taken from the test tube and diluted 40 times in a solution of acetonitrile +1% acetic acid and distilled water (1:1).

The active substance content was analyzed with a liquid chromatography-tandem mass spectrometry (LC-MS/MS) system with mobile phases of 0.1% formic acid and five mM ammonium formate in water (eluent A) or in 95% methanol, 5% water (eluent B). The gradient used to elute all compounds from the column is shown in Table S2. LC-MS/MS measurements were performed on a Xevo TQ-S (tandem quadrupole mass spectrometer) system coupled with an Acquity UPLC (ultra-performance liquid chromatography) system, both from Waters (Milford, MA, USA). Each LC-MS/MS analysis included a calibration curve of nine fortified blanks (0, 0.125, 0.25, 0.5, 1, 2.5, 5, 10 and 25 ng·mL$^{-1}$) in a solution of acetonitrile +1% acetic acid and distilled water (1:1). The extraction procedure performed on the plastic not incubated with pesticides provided a matrix extract. A standard in matrix was prepared from the matrix extract fortified at five ng·mL$^{-1}$ and injected after every 10 sample measurements. The software MassLynx$^{TM}$ (Version 4.1; Waters, Milford, MA, USA) was used to collect the data and integrate the peaks. A limit of quantification (LOQ) was calculated for each compound according to the lowest calibration level inside the linearity range (deviation of back-calculated concentration from true concentration within ±20%) and an ion ratio within ±30% of the average of calibration (*European Commission, 2017*)

(Table S4). The active substance contents below the LOQ were considered to be zero during data processing.

## Data processing

Data calculation and plotting were done in R version 3.4.2. Calculations were performed using the mean of the duplicate treatments. The percentage of sorption was calculated for each tube containing the plastic mulch and the active substances mixture as the ratio between the mass of active substances detected in the plastic extract and the mass of the active substances added. The sorption calculation did not take into account the possible decay of active substances in the solution or the incomplete extraction of sorbed substances. As a consequence, calculated sorption percentages may have been lower than the real sorption of active substances on plastic mulches. The Shapiro–Wilks test was performed to check the normality of the percentage of sorption with the function *shapiro.test* of R. Then percentages of sorption were compared among the different plastic types using a one-way non-parametric ANOVA (Kruskal–Wallis comparison with functions *kruskal.test* and *dunnTest* in R. Given $p$-values were adjusted using the Benjamini–Hochberg method.

The recovery ratio (sum of the mass of active substances measured in all compartments divided by the mass of active substances added) was calculated for each treatment where active substances were added. The difference between the mass recovered and the mass added was considered the mass of active substances decayed during the incubation. The decay calculated for active substances without plastic, minus the decay calculated in the presence of plastic, gave an estimation of the decay reduction in the case of active substance sorption.

The *lm* function of *R* calculated the linear coefficients, the coefficient of determination ($R^2$) and the $p$-value of linear regressions between percentage of sorption and log P, as well as between decay reduction and percentage of sorption.

## RESULTS

### Active substance sorption on plastic mulch

The mean sorption rate of all active substances on each type of plastic are shown in Fig. 2. The measured sorption rates did not follow a normal distribution (Shapiro–Wilks test, $p < 0.001$). LDPE and PAC plastics showed no significant differences for sorption of active substances ($p > 0.05$), with an average of ~23%. Bio mulch showed a significantly higher sorption rate than LDPE and PAC mulches ($p < 0.05$) with an average value of ~50%. In fact, 20 out of the 38 tested compounds showed a sorption rate >50% on Bio, whereas only nine and seven had a sorption rate >50% on LDPE and PAC mulches, respectively.

Sorption rates (%) of each active substance to the different plastic types (treatments LDPE+P, PAC+P and Bio+P) are presented in Fig. 3 and in Table S4. We observed that 14 compounds (37%) had a sorption rate >10% on all plastic mulches. Ten compounds had a sorption rate <1% on all plastics. Three compounds, Chlorpyrifos, Oxyfluorfen and Pendimethalin, had a sorption rate >80% on all plastics.

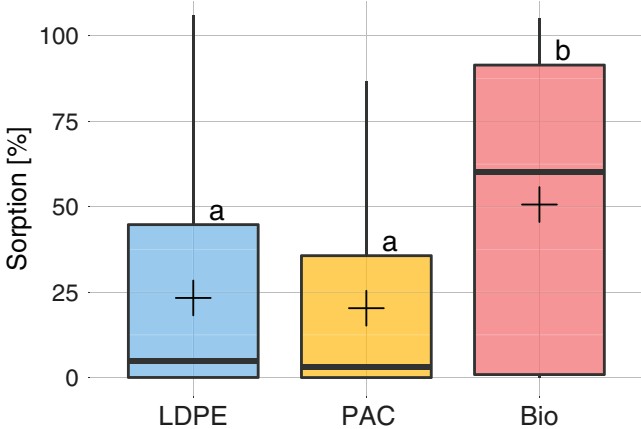

**Figure 2 Sorption (%) on each type of plastic: LDPE (blue), PAC (orange) and Biodegradable (red).** The box plot (horizontal lines) represents sorption for at least 25%, 50% and 75% of the active substances. The vertical black line ends represent the minimum and maximum values. The cross is the mean sorption for all active substances. Different letters indicate significant differences among plastic types after a Kruskal–Wallis comparison at $p < 0.05$.

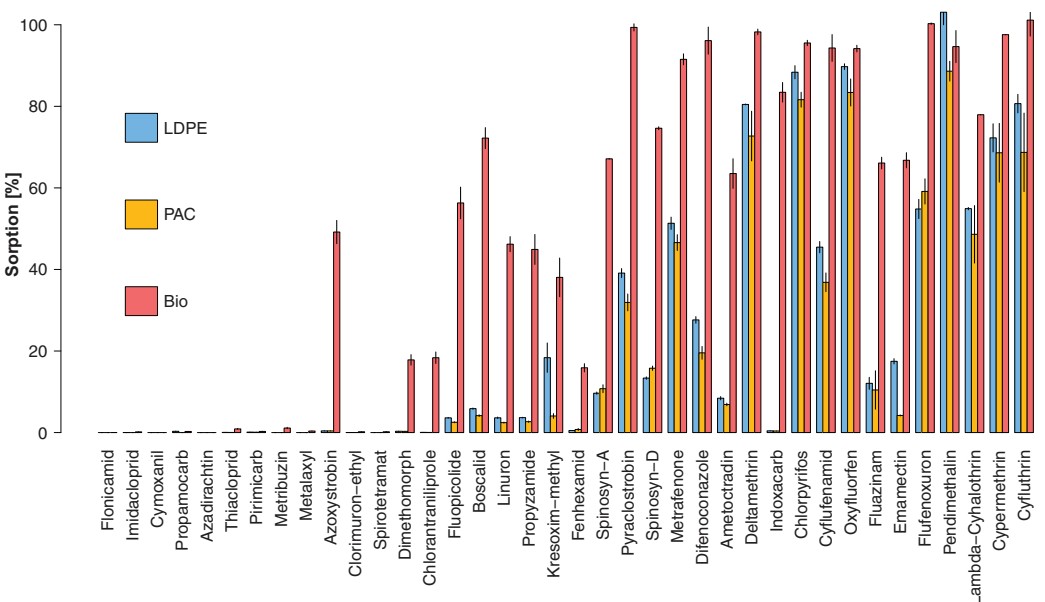

**Figure 3 Mean sorption rate (%) for each active substance on LDPE (blue), PAC (orange) and Biodegradable (red) plastic mulch.** Vertical black lines represent the measurement ranges (min and max). Active substances are ordered according to increasing log P (octanol-water partition coefficient).

Active substances with higher log P values tended to show higher sorption rates. In fact, the sorption rate was positively correlated with log P ($R^2 = 0.96$, $p < 0.001$) (Fig. 4). Nevertheless, it is worth noting that active substances with the same log P and the same plastic type could have had significantly different sorption rates (e.g., Fig. 3; log P (Chlorpyrifos) = log P(Cyflufenamid) = 4.7 whereas mean sorption on LDPE was 88% for Chlorpyrifos and 45% for Cyflufenamid).

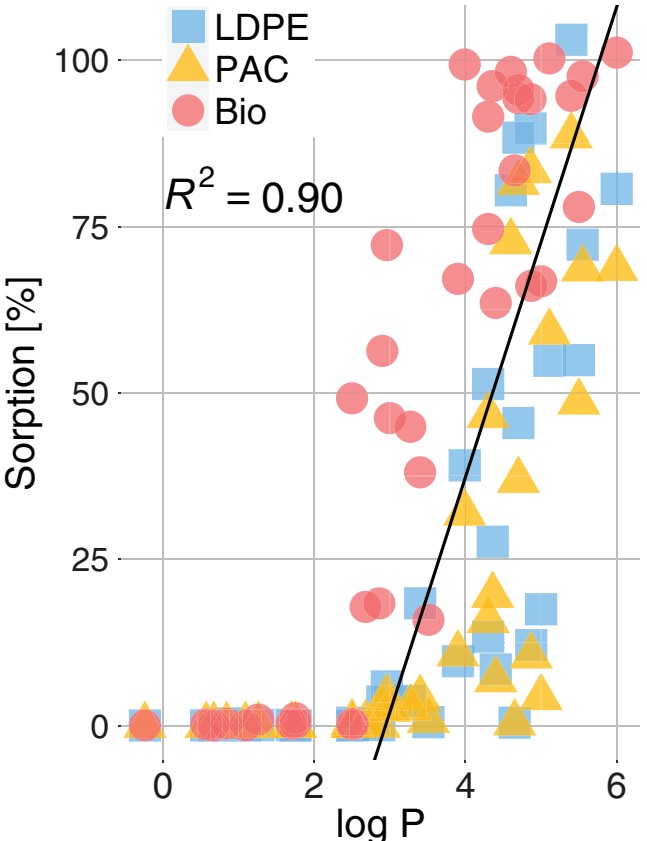

**Figure 4  Sorption (%) of active substances on LDPE, PAC and Bio mulch as a function of the active substances log P (octanol-water partition coefficient).** The black line is a regression calculated for sorption >0: $y = 35x - 104$; $R^2 = 0.93$; $p < 0.001$.

For the plastic mulch with no added active substances (LDPE+W, PAC+W and Bio+W), no active substances were found in the PAC or Bio extracts but low levels (maximum: 16 ng; average: 7.7 ng) were found in the samples with LDPE in one duplicate but not in the other duplicate. These very low levels (compared to the 5,000 ng of active substance added) are likely to have come from contamination in the liquid chromatography column.

For active substance samples where no plastic was added, half of the compounds had a recovery ratio <50% and 29 compounds (75%) had a recovery ratio <90% (Fig. S2). It is worth noting that without plastic, active substances with higher log P tended to have lower recovery. Moreover, active substances with a shorter DT50 in water tended to have a lower recovery ratio; meaning that lower recovery is likely to be explained by higher decay during incubation. In the next section, we assumed that the missing percentage (1-recovery) was due to the degradation of the active substances during incubation.

## Decay reduction due to sorption of active substances

The sorption of active substances significantly reduced their decay in comparison to the active substances without plastic mulch (Fig. 5). The estimated decay for active substances

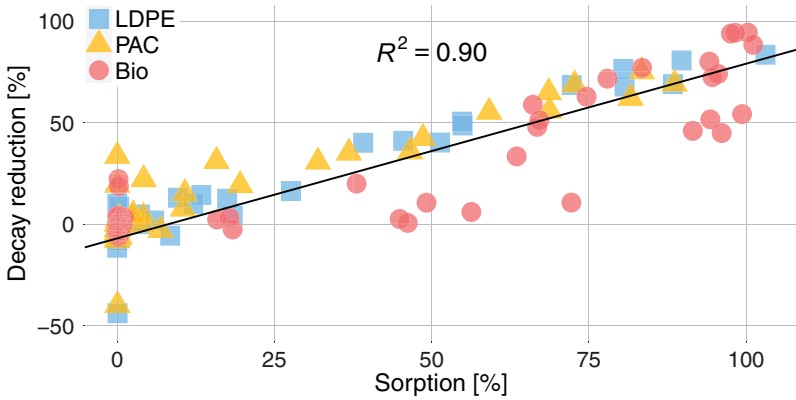

**Figure 5 Decay reduction (%) for active substances sorbed on plastics related to the sorption (%) of active substances for the three types of plastic, LDPE (blue square), PAC (orange triangle) and Biodegradable (red circle).** The decay reduction (%) is the difference between the decay of an active substance in presence of plastic and the decay of the same active substances without plastic. The black line is the regression calculated for sorption >0: $y = 0.86x - 7$; $R^2 = 0.90$; $p < 0.001$.

with sorption >80% was ~70% lower than the decay without plastics. We measured a decay reduction of ~27% for LDPE and PAC and ~37% for Bio for active substances with a sorption >0.01%. The decay reduction showed a significant linear relationship with the percentage of sorption ($R^2 = 0.90$, $p < 0.001$). The greater sorption on Bio directly reflected on a greater decay reduction for active substances.

## DISCUSSION

We did a single point sorption experiment. Based on previous studies, 15 days of sorption at 35 °C were enough to reach steady states (*Nerín et al., 1996*; *Sharom & Solomon, 1981*). Kinetic sorption experiments would be needed to calculate sorption coefficients. The active substance extraction procedure was partly based on *Nerín et al. (1996)* and was not tested again. In case of a low extraction rate, we underestimated the sorption on plastic and over-estimated the decay of active substance incubated with plastic. We don't expect the extraction rate to perform differently for LDPE, PAC or Bio. Therefore, a low extraction rate would not change our conclusion.

The Bio mulch may have contained polybutylene adipate terephthalate and Bio mulch had a higher sorption than LDPE and PAC mulches. Higher sorption may be related to the chemical properties of the polymer used or to the specific surface area of the mulch (*Aslam et al., 2013*). Polybutylene adipate terephthalate may have a better affinity with the active substances than the LDPE because of aromatic-interactions and potential hydrogen bonds when the polymer is altered (*Palsikowski et al., 2018*). Aged polybutylene adipate terephthalate is able to form hydrogen bonds with organic chemicals (*Weng et al., 2013*). It is possible that the 15 day incubation at 35 °C caused an alteration in the Bio mulch and increased the formation of hydrogen bonds between the Bio mulch and the active substances. Additionally, biodegradable mulches tend to be made with smaller polymer fiber diameters that increase its specific surface area (*Chinaglia, Tosin & Degli-Innocenti, 2018*;

*Hayes et al., 2012*) and its biodegradation (*Brodhagen et al., 2015*; *Chinaglia, Tosin & Degli-Innocenti, 2018*). A greater specific surface area would also increase the sorption (*Liu et al., 2019*).

Our study showed that for 20 compounds sorption on Bio mulch was higher than 50%. Sorbed active substances are likely to alter the degradation of the Bio mulch by affecting the soil microbiome (*Oyeleke & Oyewole, 2019*). According to the International Organization for Standardization 17556, biodegradable mulch should reach at least 90% biodegradation in the soil within 2 years (*Carol et al., 2017*). A field study showed that after 397 days in the soil, three different kinds of biodegradable mulches (Crown 1, BioAgri and SB-PLA-11) had various deterioration rates (100%, 65% and very little deterioration, respectively) (*Cowan, Inglis & Miles, 2013*). Plastic degradation studies should take into account that pesticides are likely to be sorbed on plastic and may reduce its biodegradation. On the other hand, the efficiency of pesticides in the soil (i.e., herbicides, fungicides) depends on their availability. Therefore, plastic mulch may decrease the efficiency of pesticides by decreasing their release into the soil when plastic mulch debris accumulates and the pesticides are sprayed on the soil. The aging of plastics (*Liu et al., 2019*) and the pesticide sorption in soils contaminated with plastics (*Hüffer et al., 2019*), for different soil types and organic matter contents, are factors that need to be studied to understand the interactions between plastic debris and pesticide residues.

Low Density Polyethylene and PAC mulches have similar sorption and their composition only differs in the additives that are added to them. Thus, the additives present in PAC mulch do not seem to change the sorption property of the original LDPE polymer. The sorption was much higher for the Bio mulch. As a consequence, knowing the exact chemical formulation of the polymers used to make plastic mulches and the specific surface area of the mulch are essential in understanding the mechanisms of the sorption of pesticides on plastic. Better, cheaper and faster analysis of plastic composition (*Corradini et al., 2019*; *Mintenig et al., 2017*) or regulations forcing producers to share the chemical formulation of polymers could help filling this knowledge gap.

The sorption on plastic varied for all active substances, being higher for those with higher log P. In fact, log P, as a measure of hydrophobicity (*Leo, Hansch & Elkins, 1971*), plays a key role in sorption processes (*Aslam et al., 2013*). Nevertheless, the log P did not predict the sorption for all active substances so mechanisms other than hydrophobicity must play a role (*Guo et al., 2000*). Some active substances may have had an impact on the sorption of some other ones, likely due to chemical interactions between them. Interactions between active substances might have changed the active substance degradation in solution as well as in the plastic matrix. These results highlight the need for more detailed studies to understand the mechanisms of pesticide sorption on plastic.

The highest degradation rates were obtained for active substances with low stability to hydrolysis and low stability in aqueous solution (aqueous hydrolysis DT50 (days) at 20 °C, pH 7 and degradation in water DT50 (days) (*PPDB, 2018*)). We can assume that the pesticide degradation in the glass tube was mainly due to hydrolysis (*Fenner et al., 2013*). Volatilization in the gaseous phase in the tube or incomplete solubilization could have played a role in the estimation of the decay. However, neither the decay reduction nor the

sorption was correlated with the solubility, meaning that hydrolysis was the most likely process leading to the degradation. The decay of active substances from pesticides was reduced by sorption. It is commonly accepted that sorption limits pesticide degradation by reducing its partitioning into the liquid phase (*Guerin & Boyd, 1997*, *O'Loughlin, Traina & Sims, 2000*). Additionally, soil microorganisms degrade preferably or exclusively chemicals that are present in the soil solution (*Boivin, Cherrier & Schiavon, 2005*). Thus, sorbed active substances would undergo less degradation by microorganisms (*Liang et al., 2011*). Since plastic debris could be transported by wind and water (*Liu, He & Yan, 2014*), pesticide transport (*Teuten et al., 2007*) and degradation models (*Silva et al., 2018b*) should take into account the sorption of pesticides on plastic (*Villeneuve et al., 1988*).

The applied experimental design was based on *Nerín et al. (1996)* to reveal a potential of commonly used active substances from pesticides to be sorbed and protected from decay on conventional (LDPE) and new (PAC and Bio) plastic mulches. The sorption condition applied here does not reflect real conditions in fields. The 38 active substances were applied together at the same concentration, in the same solution. We can assume then that most hydrophobic active substances may have reduced the sorption of the rest of the substances due to the competitive sorption among all active substances. Competitive sorption would occur to a negligible extent in the field because fewer active substances would be applied simultaneously and the use of pesticides would be spread over the whole growing period resulting in seasonal variations. Moreover, the applied concentration of 1,000 ng·mL$^{-1}$ exceeded the solubility in water for some compounds. A non-dissolved fraction of the active substances may had formed a stock within the liquid medium. The incubation was done in glass tubes, in the dark, at 35 °C without temperature variation, stirring or oxygenation. In the field, active substances could undergo sorption on soil particles, degradation by light and by microorganisms or volatilization. Additionally, the presence of 10% acetonitrile in the incubation solution could have reduced the sorption of most hydrophobic contaminants in plastics and polymers (*Teuten et al., 2007*). Sorption percentages on plastic may be higher without acetonitrile and with less competitive sorption. However, sorption percentages could be reduced by additional degradation processes (*Fenner et al., 2013*; *Kumar et al., 2018*), volatilization and sorption to soil particles (*Boivin, Cherrier & Schiavon, 2005*). Finally, plastic degradation may change the chemical properties of the polymer or the specific surface area of the mulch (*Hayes et al., 2017*; *Liu et al., 2019*) and change the sorption of active substances (*Aslam et al., 2013*). Despite these last issues comparing the conditions of this experiment with actual conditions in the field, our study highlights the need for further research on plastic mulch-pesticide systems since there is a real interaction between both components, which could negatively affect pesticides and plastic degradation in the field. Thus, it is essential to address these topics under field conditions (*Yang et al., 2018*) taking into account new and aged plastics (*Liu et al., 2019*).

The sorption of active substances on plastic may change the toxicity of both the pesticides and the plastic. In fact, if plastic debris is ingested by organisms (*Colabuono, Taniguchi & Montone, 2010*; *Huerta Lwanga et al., 2017*) then the sorbed active substances

could potentially be desorbed in organisms (*Teuten et al., 2007*). On the other hand, the sorption to plastic may reduce the bioavailability of active substances, especially reducing the peak concentration after pesticides application. The reduced exposure of soil organisms could be beneficial for the ecosystem. Contaminated plastics may as well release active substances in the soil solution and contribute to plastic toxicity (*Machado et al., 2018*; *Qi et al., 2018*). In a similar way, active substances sorbed on plastic may decrease the plastic's degradation by soil organisms because of the toxicity of the active substances. Active substance sorption is of particular concern when it comes to the degradation of biodegradable mulch since the degradation of biodegradable mulch relies heavily on the activity of microorganisms in the soil. As a consequence, further studies are needed to assess the degradation of plastic debris, especially biodegradable plastics in soils where pesticides are sprayed.

## CONCLUSIONS

This study reveals that sorption of active substances on plastic depends on both the chemical structure of the active substance and the type of plastic mulch. Sorption was higher for active substances with higher log P (octanol-water partition coefficient) and although it was similar between LDPE and PAC plastics, it was significantly higher on Bio mulch. Moreover, sorption of active substances on plastic reduced the decay of active substances. Therefore, the sorption of active substances can change the eco-toxicity and decay of both the active substances and the plastic debris. The sorption can also affect the transport pattern of active substances, especially when biodegradable plastic is used. More research is needed to evaluate the dynamics and consequences of the sorption of active substances from pesticides on plastic mulches in environmental conditions. With more research, scientists can propose guidelines for the use of plastic mulches in agro-ecosystems in order to avoid soil and water pollution.

## ACKNOWLEDGEMENTS

We are thankful for the contribution of the farmers from the region of Murcia, Spain. We would like to thank Klaas Oostindie for his graphic support and Robin Palmer for the language editing. We are very grateful to Frits Gillissen from the Aquatic Ecology and Water Quality Management group of Wageningen University & Research for the FTIR analysis of the plastic mulch. We would also like to thank Esperanza Huerta Lwanga for her invaluable comments and revisions.

### Funding

This work was supported by the European Commission Horizon 2020 project Diverfarming (grant agreement 728003). The funders had no role in study design, data collection and analysis, decision to publish, or preparation of the manuscript.

## Grant Disclosures

The following grant information was disclosed by the authors:
European Commission Horizon 2020 project Diverfarming: 728003.

## Competing Interests

Violette Geissen is an Academic Editor for PeerJ.

## Author Contributions

- Nicolas Beriot conceived and designed the experiments, performed the experiments, analyzed the data, prepared figures and/or tables, authored or reviewed drafts of the paper, and approved the final draft.
- Paul Zomer conceived and designed the experiments, performed the experiments, analyzed the data, authored or reviewed drafts of the paper, and approved the final draft.
- Raul Zornoza conceived and designed the experiments, authored or reviewed drafts of the paper, and approved the final draft.
- Violette Geissen analyzed the data, authored or reviewed drafts of the paper, and approved the final draft.

## Data Availability

The raw data from the LC-MS/MS analysis are available in the Supplemental Files.

## Supplemental Information

Supplemental information for this article can be found online at http://dx.doi.org/10.7717/peerj.9876#supplemental-information.

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
