# Peer review of "A laboratory comparison of the interactions between three plastic mulch types and 38 active substances found in pesticides"

_PeerJ, doi:10.7717/peerj.9876_

## Round 0.1 · original submission · Major Revisions

Our apologies for the delay in coming back to you. It has taken longer than expected to find adequate reviewers. I am now confident that both reviewers have provided a detailed review of your manuscript and made pertinent suggestions for improvements that you should consider.

Reviewer 1 ·

Basic reporting

The manuscript is well structured, with relevant figures and tables. It is self-contained with relevant results and hypotheses.

The English language is in general comprehensible and easy to read. There are several grammar mistakes throughout the manuscript though that need to be corrected. Some examples include lines 25 (no data is), 38 (assess __ behaviour), 48 (in __ European Union), 52 (debris are left), 66 (debris are more), etc. These are only some examples; please consider getting the full manuscript revised by an English native speaker.

Please review the way citations are presented: it is common practice that only the first author name is referred followed by “et al” when there are more than 2 authors – check for example lines 51, 57, 64, 67, etc. Also, when a sentence starts with a citation, it should be referred the author(s) name followed by “et al” (if more than 2 authors) and followed by the year of the publication in brackets; the authors of this manuscript aren’t following this rule – see for example lines 93 and 94: Nerin et al. (Nerín, Tornés et al. 1996) and Sharom and Solomon (Sharom and Solomon 1981). The full manuscript needs to be revised for these situations.

Experimental design

The manuscript presents a laboratory experiment where the sorption of 38 active substances of pesticides into 3 types of plastic mulch is being tested. Additionally, the authors investigated the decay of the active substances in the presence/absence of plastic mulch.
The Experimental design has been properly designed and performed and it is of relevance to the scientific community as well as to society and policy makers in general. The methods were well described and, despite minor issues, they have sufficient detail and information to be replicated by others.

I do have some questions/remarks/comments though:
1. As described by the authors in the introduction and further discussed in the discussion chapter, plastic mulch is used in agriculture and therefore it is mostly in contact with soil. Moreover, most of the used pesticides are applied in the soil, that then will be in contact with the plastic; or they are applied post emergence in the crops, reaching the plastic mulch and most probably the soil as well. This means we are in the presence of a solid(soil)-solid(plastic mulch) phase and not a liquid(incubation solution)-solid(plastic mulch) phase as performed by the authors.
1.1 How does the performed experiment in a liquid-solid phase translate to the reality of a solid-solid phase? How relevant is this experiment in a liquid-solid phase to the reality observed in the field?
1.2 Why didn’t the authors test with soil (solid-solid), which would represent better the reality?

2. It is shortly described the procedure for the extraction from the plastic pieces, but it isn’t described the procedure for the incubation solution. Was the incubation solution directly analysed without any extraction? Even if so, the authors had to include IS correct? Please give a short description on how was the incubation solution prepared for analysis in the LC-MSMS.

3. The authors followed the method of analysis of Mol et al (2008) – analysis for food/feed, and Silva et al (2018) – analysis in soils. How certain are the authors about the good extraction efficiency from the plastics? Were tests performed in plastic to verify the good extraction procedures and to obtain the recoveries from plastic? It is a different matrix from soils or food/feed, adsorption is likely to behave differently and therefore extraction might be easier or more difficult.

Ls 221 & 223: Please give the full names of the acronyms used when used for the first time (LC-MS/MS; TQ-S; UPLC)

L225: The blanks and standards were prepared in Milli-Q water? Please specify.

Ls 227 to 230: From the authors’ description, and assuming that the calibration standards were prepared with Milli-Q, it means the LOQ is based on the best scenario analysis, meaning it is based on the best signals you can obtain for your analysis. Matrix effects from the plastics extraction may have happened though. Didn’t the authors test for these possible matrix effects? How did the authors guarantee that the LOQ isn’t at a higher level due to these effects?

Validity of the findings

Conclusions are well stated and linked to the original research questions and are somehow limited to supporting results.
All data in which the conclusions are based have been provided.
The manuscript presents enough impact and novelty to be accepted for publication after major revisions.

Additional comments

1. Consider using the term “adsorption” instead of “sorption” throughout the manuscript when referring to the adsorption of pesticides into the plastic.

2. Say what is the incubation solution composed of (10% ACN, 90% H2O) already in the abstract and also the first time you refer to it in the methods section (in L142). It isn’t nice nor good to the reader to have to wait until section “Incubation solution” to know what is the solution composed of.

3. Throughout the whole manuscript, consider substituting “square” (when referring to the piece of plastic) by “piece” or similar to avoid confusion with the square from cm2.

4. I couldn’t find any legends for the figures and tables in supplementary material, please include such.

L32: consider removing the word “square” – cm2 already has the “square” meaning on it.

L74: remove the space after “:” and lowercase the “T” of “the”.

L113: The sentence starting with “As a preliminary investigation…” should be a new paragraph.

L142: remove “a)”

L173: remove “squares”

L174, L201, L370: a sentence should not start with numbers (“3x3”, “15”, “38), rephrase the sentences accordingly.

L208: make it “quantification”

L287: use normal brackets for the presented number of samples.

L287: do the authors by “ lower DT50” mean “shorter DT50” or “longer DT50”? Consider changing to shorter or longer.

Ls 292 to297: This section on the decay reduction should be a little bit more developed. Only a comparison is made between the decay without plastics and with plastics in general. But could there be a plastic type that could affect the decay more strongly than another type? From the results presented in Fig. 3, a greater adsorption was observed for Bio plastics – did this reflect a greater decay reduction for active substances with Bio plastics as well? Consider changing Fig. 5 to show the decay reduction for each type of plastic and verify as well if there are significant differences in decay reduction between plastic types.

L301: Include references for the previous studies referred.

Ls 311-312: The authors refer that the time of incubation and temperature may have increased the adsorption to the Bio mulch due to the formation of H-bonds. This might also be true due to the fact the experiment was performed in a liquid-solid (incub. solution-plastic) phase. Would the authors expect the same to happen in reality, with a solid-solid (soil-plastic) phase?

L329: … “the pesticides sorption in soils”… Do the authors really mean “in soils” here? In general, adsorption of pesticides in soil is known. I believe the authors mean to refer something related to the plastics instead? Please revise.

L330: remove the “s” from “soils”

L350-353: The authors analysed the samples only once, 15 days after incubation (not at time 0 or at different time periods), so how certain and confident are the authors that the referred “decay” wasn’t linked to a poor extraction procedure due to strong sorption to the plastics of some compounds?

L357-366: “The decay of active substances from pesticides was supposed to be reduced by sorption.” Why are the authors saying “supposed to be”? The results presented by the authors proved that this effectively happened (assuming a good extraction performance of the method of analysis)… Please consider revising this sentence and the whole paragraph for stronger statements and based on your results. The way it is written leaves the feeling that this was not what happened in your results, which isn’t true. This whole paragraph refers too much uncertainty when studies already proved most of this (“commonly accepted”, “plastic debris may reduce”, etc).

L377: “A not-dissolved fractions” - or you remove the “s” from “fractions”, or you remove the “A” at the beginning.

FIG.1 – There is a typo in the legend: in the 3rd description it says “soultion” instead of “solution”. Please correct it. Also, again, remove the “square” and substitute it by “plastic piece” or similar. Moreover, include information on what is the solution composed of.

Table 1: include what is the incubation solution composed of.

Table S2: Include the reference(s) for the information you provide.

Table S3: Please include SD for sorption and decay results. Maybe due to the lack of a legend to the table, it isn’t comprehensible what the results presented really mean: sorption and decay after 15 days? For example, azadirachtin is shown with a decay of 100% - does this mean that this compound degraded completely? You also present negative percentages (ex: Clorimuron-ethyl, linuron, etc) – what do these negative values mean? Please find a way of making these aspects comprehensible by reviewing the titles of the columns for example. The table needs to be able to stand alone.

Reviewer 2 ·

Basic reporting

The title requires re-structuring. In its present for, it suggests that the tests were conducted in three types of mulch. Instead, it was in three types of plastic using in mulching. Please rephrase. Some minor phrase corrections are needed, such as in the Introduction, where it should read, for example, “..year in the European Union” or “debris leads to its accumulation”, or “A laboratory single point sorption experiment was”. However, these are minor issues that do not affect the readability of the manuscript, although proof-reading is recommended.
Discussion
“Additionally, sorbed chemicals are generally assumed to be less accessible to microorganisms,
which preferentially or exclusively utilize chemicals in solution” – Please clarify.

Figure 2 – I do not understand exactly how what the authors sate in the caption relate exactly with the image, regarding the vertical black lines. Is this an interval, or are these error bars?
I believe that some listed references that are not cited in the text (e.g., Besseling et al, 2017), but I may be mistaken. Please verify.

Experimental design

The main limitation of the work stems from the fact that no kinetics can be deduced from the provided data, as the authors made single-point measurements. Although this may be understandable, taking into consideration the large number of pesticides tested, this work would have been of immensely larger significance had the authors opted to study these effects on a smaller set of representative pesticides, but included multiple measurement points. Additionally, these tests could have been conducted resorting to environmentally significant concentrations of these pesticides. The authors were careful to include types of plastics and pesticides used in intensive agricultural activities, but the concentrations used, as the authors themselves noted, are not environmentally relevant. There are also some methodological issues that need further elucidation. For example, why was an ultrasonic bath used? This certainly does not mimic real-world settings. Also, the LC-MS/MS method needs to be better described, regarding the elution conditions. Was this carried out in gradient, or isocratic mode? In the case of the former, please provide the elution profile. In the case of the latter, please provide ratios. I certainly appreciate the authors providing the LC-MS/MS raw data, but it would have been nice to see at least an example of a typically obtained mass spectrum and LC chromatogram. Furthermore, was the described method validated? If so, data for this need to be provided. Was there any mass balance conducted, that quantifies the pesticides found adsorbed on the plastics and those that remained in the solution? This should be done.

Materials and Methods.
Please include number of scans for FTIR and the background.
3×3 cm² squares/15 days/38 – Please do not begin a sentence with a numeral.
Equipment, such as incubators, ovens, etc, need to be identified (make and model).

Validity of the findings

Please refer to other sections, in which the methodological approach is assessed as well as the associated validity issues.

Additional comments

Beriot and colleagues have submitted a manuscript in which they compare the putative interactions between three types of plastic materials and numerous pesticides. Although certainly interesting, the work carried out by the authors requires some clarifications.

---

## Round 0.2 · Minor Revisions

Thank you for considering the reviewers' comments in details. Please consider further suggestions for minor changes.

Reviewer 1 ·

Basic reporting

The revised manuscript has improved its quality. There are minor mistakes that still need to be corrected though.
I annotated my comments in the uploaded word document after the responses from the authors (in red, for Reviewer 1) - document named "peerj-46097-Rebuttal_Letter_04_06_2020_R1".
I also highlighted most of the necessary changes in the uploaded pdf - document named "peerj-reviewing-46097-v1_R1".

Experimental design

No comment

Validity of the findings

No comment

Annotated reviews are not available for download in order to protect the identity of reviewers who chose to remain anonymous.

---

## Round 0.3 · accepted · Accept

We thank you for considering the reviewer´s comments and further revising the manuscript